# Molecular Tailored Therapeutic Options for Advanced Gastrointestinal Stromal Tumors (GISTs): Current Practice and Future Perspectives

**DOI:** 10.3390/cancers15072074

**Published:** 2023-03-30

**Authors:** Fabio Catalano, Malvina Cremante, Bruna Dalmasso, Chiara Pirrone, Agostina Lagodin D’Amato, Massimiliano Grassi, Danila Comandini

**Affiliations:** 1Medical Oncology Unit 1, IRCCS Ospedale Policlinico San Martino, 16132 Genoa, Italy; 2Genetica dei Tumori Rari, IRCCS Ospedale Policlinico San Martino, 16132 Genoa, Italy

**Keywords:** advanced GISTs, clinical practice, molecular characterization, clinical trials, precision oncology

## Abstract

**Simple Summary:**

GISTs are the most common mesenchymal tumors of the gastrointestinal tract. This review provides an accurate and in-depth rundown on the molecular pathways that characterize gastrointestinal stromal tumors (GISTs), together with the state of the art and future perspectives of the tailored treatment strategies studied and under investigation for this disease. The first part of this review may be a useful tool for all clinicians facing this disease in their daily clinical practice. In addition, the last chapters explore new treatment options, demonstrating encouraging preliminary results that may become the standard of care in the future.

**Abstract:**

Gastrointestinal stromal tumors (GISTs) are one of the most common mesenchymal tumors characterized by different molecular alterations that lead to specific clinical presentations and behaviors. In the last twenty years, thanks to the discovery of these mutations, several new treatment options have emerged. This review provides an extensive overview of GISTs’ molecular pathways and their respective tailored therapeutic strategies. Furthermore, current treatment strategies under investigation and future perspectives are analyzed and discussed.

## 1. Introduction

Gastrointestinal stromal tumors (GISTs) are the most common mesenchymal tumors of the gastrointestinal (GI) tract, accounting for 80% of all GI mesenchymal tumors and 0.1 to 3% of all GI malignancies [1]. In 2000, GISTs were formally recognized as a specific subtype of neoplasm, and their incidence has been increasing steadily with improvements in diagnostic technologies [2].

To date, the annual incidence of GISTs ranges from 10 to 15 cases per million [3]. Most GISTs occur in middle-aged adults (median age of 60–65 years), without gender predominance [3,4].

However, a minority can be found among children and young adults. In particular, in children, they may arise as part of the non-hereditary Carney triad or autosomal-dominant Carney–Stratakis syndrome, with predisposing germline SDH subunit mutations [5,6] or as part of neurofibromatosis 1 [7].

GISTs usually arise as a submucosal or subserosal mass in the stomach (60%), the small intestine (25%), or, less commonly (<3%), in the colon, the rectum, the esophagus, the mesentery, or the omentum [8]. About 47% of GISTs are diagnosed at an advanced stage, usually with metastases to the liver and peritoneum [9].

GISTs may harbor specific mutations in typical genes. About 75–80% of GISTs have mutations in the KIT gene, while 5–10% of them have mutations in the platelet-derived growth factor receptor a (PDGFRA) gene [10]. Around 15% of cases, may be driven by other different mechanisms, which may include the inactivation of the NF1 gene, inactivation of genes encoding succinate dehydrogenase (SDH) subunits, or the molecular alterations of genes, such as RAS, BRAF, PIK3CA, ETV6, NTRK3, and FGFR1. Nevertheless, a small amount of GISTs, to date, have not been characterized by any driver mutation and are therefore classified as wild-type GISTs [11,12,13].

Despite their propensity for multifocality and lymphatic spread, SDH-deficient GISTs usually follow a more indolent clinical course compared to KIT/PDGFRA-mutant GISTs [14].

The standard treatment of localized GISTs is complete surgical resection; nonetheless, the relapse rate remains around 50% [15]. Both the NCCN and ESMO guidelines, as well as the consensus of the scientific community, recommend 3 years of adjuvant treatment with imatinib in high-risk patients [16]. TKIs other than imatinib are not indicated as adjuvant treatment [9,17,18].

On the other hand, the treatment for metastatic or unresectable GISTs includes different TKIs, such as imatinib, sunitinib, regorafenib, ripretinib, and avapritinib. In advanced settings, the median overall survival has improved from 18 months to more than 70 months since the introduction of these drugs [19].

The aim of this review is to provide an overview of molecular-tailored treatment options for advanced GISTs.

## 2. Diagnosis and Molecular Characterization

GIST diagnosis usually follows the discovery of a mass of apparent gastrointestinal origin, either due to clinical manifestations, such as gastrointestinal bleeding and epigastric discomfort, or due to an accidental discovery during an imaging exam performed for unrelated reasons. Following medical history and physical examination, initial diagnostic approaches include an abdominal/pelvic CT scan and/or RMN, especially in the case of rectal GIST, where the level of anatomical detail provided by RMN allows for more accurate preoperative staging and is useful for conservative surgery planning. A complementary chest CT scan is recommended for staging purposes. In individuals eligible for targeted therapies, such as imatinib mesylate, FDG-PET can be useful for the purpose of monitoring the early response. Histopathological evaluation of a tumor sample is recommended, especially in the case of lesions >2 cm, which are at higher risk of progression [17].

### 2.1. Diagnostic Biomarkers

In addition to the morphological characterization of bioptic samples, molecular biomarkers are employed for GIST diagnosis. KIT (CD117) is a commonly used diagnostic biomarker, as nearly all GISTs (>95%) show positive CD117 immunohistochemistry (IHC) staining [20]. KIT IHC positivity is independent of underlying somatic *KIT* mutations, as it can be found in *KIT* wild-type tumors, but, on the other hand, it can be absent in a subset of *KIT*-mutated tumors. In recent years, anoctamin-1 (DOG1) has emerged as an additional diagnostic biomarker with higher sensitivity than CD117, as it can detect up to 35% CD117-negative GISTs. Unfortunately, high CD117 and DOG1 sensitivity is not matched by an analogous specificity, as both can also be expressed by other cancer types [21].

In addition to CD117 and/or DOG1, the majority of gastric GIST, and around half of those arising at other sites express the cluster of differentiation 34 (CD34) antigen. Smooth muscle actin (SMA) expression can be detected in up to 25% of GISTs. Conversely, two other molecules, protein S100 and desmin, are expressed by no more than 10% of GISTs [21,22]. SDH immunohistochemistry can be useful to identify a subset of *KIT/PDGFRA* WT GISTs.

### 2.2. GIST Molecular Subtypes

GISTs can be classified into molecular subtypes according to a subset of recurrent genomic aberrations.

The most frequently altered gene is *KIT* [20], followed by *PDGFRA* [22,23,24]. Within the other rarer molecular subtypes, almost half of them are characterized by mutations or epigenetic silencing of one of the succinate dehydrogenases (SDH) complex genes, whereas a smaller subset harbors *BRAF* or *NF1* mutations. Molecular characterization of GISTs is crucial for therapy decision-making, as specific mutations determine one’s eligibility for treatment with tyrosine kinase inhibitors (TKI) and/or drug dosage [25,26]. Moreover, both preclinical studies and clinical trials are currently investigating potential additional therapies according to the GIST mutational profile. In addition to its therapeutic implications, molecular analysis can integrate the diagnostic process in specific situations; for instance, the identification of a *KIT* or *PDGFRA* mutation in a suspected GIST with negative CD117 and DOG1 expressions could aid differential diagnosis [17].

#### 2.2.1. KIT

*KIT* encodes the KIT proto-oncogene, a transmembrane protein acting as receptor tyrosine kinase. c-KIT belongs to the type III receptor tyrosine kinase family, which also includes PDGFRA, PDGFRB, CSF1R, and FLT3. c-KIT protein can be a membrane-bound or soluble protein. The interaction with its ligand SCF (stem cell factor) causes KIT activation by homodimerization and autophosphorylation [27]. KIT is involved in several pathways, including the MAPK and PI3K-mTOR signaling networks. Upon activation by its cytokine ligand, SCF, KIT phosphorylates multiple intracellular proteins involved in proliferation, differentiation, migration, and apoptosis. Activating mutations in the *KIT* gene are drivers of several tumors, including around 75% of sporadic GISTs [28]. In addition, a small fraction of hereditary GIST is characterized by germline *KIT* pathogenic variants [29].

The majority of *KIT* mutations involve either exon 11 (>75% of GISTs) or exon 9 (8–10% of GISTs).

*KIT* exon 11 mutations alter the juxta membrane domain, resulting in the disruption of its autoinhibitory activity on the kinase domain [30], thus leading to c-kit constitutional activation in a ligand-independent fashion [31,32].

The type and location of *KIT* exon 11 mutations are linked to specific tumor phenotypes. Indeed, exon 11 duplications or single-nucleotide variations are associated with a more favorable prognosis in terms of survival, whereas exon 11 deletions are associated with more aggressive tumor behavior, as well as with liver metastasis [33,34,35]. Mutations resulting in the deletion of either codon 557 or 558, for instance, were associated with the development of liver metastasis, possibly through CXCR4 upregulation [36].

Mutations located in *KIT* exon 9 alter the D5 extracellular immunoglobulin-like domain, resulting in increased sensitivity to SCF binding and increased KIT dimerization and activation [32,37,38]. Seldomly, *KIT* mutation can arise in other parts of the gene. Specifically, mutations involving exons 13, 14, and 17 have been described in about 3% of newly diagnosed GISTs [39,40]. Exon 13 mutations affect the KIT ATP-binding pocket, i.e., the first tyrosine kinase (TKI) domain, whereas exon 17 mutations, which alter the activation loop (second TKI domain), prevent KIT from reverting to its inactive form [39].

#### 2.2.2. PDGFRA

Between 10 and 14% of GISTs harbor an oncogenic mutation in the platelet-derived growth factor receptor α gene (*PDGFRA*) [41]. PDGFRA is a receptor for PDGFA, PDGFB, and PDGFC and has pleiotropic functions, including cell proliferation and migration, angiogenesis, bone marrow differentiation, and the development of the gastrointestinal mucosa. PDGFRA, like KIT, is a member of the tyrosine kinase III family. The most frequent *PDGFRA* mutations in GISTs are in exon 18, with D842V being found in up to two-thirds of *PDGFRA*-mutant GISTs. This mutation alters the protein’s activation loop and, similarly to *KIT* exon 11 mutations, this results in a constitutive activation of PDGFRA [42]. According to recent research, GISTs harboring D842V show a distinctive gene expression profile suggestive of a possible neural differentiation. Moreover, compared to GISTs with KIT or other *PDGFRA* mutations, these tumors show an enrichment of immune system/inflammation pathways and appear to have a marked immunogenicity, possibly through an increased production of neoantigens [43]. Other activating mutations can occur, less frequently, in exons 12 and 14 [23,44]. Although the majority of *PDGFRA* mutations occur at the somatic level, germline *PDGFRA* pathogenic variants cause a rare hereditary syndrome characterized by an increased susceptibility to multiple gastrointestinal mesenchymal tumors, including GISTs [45]. Due to the lack of interstitial Cajal cell (ICC) hyperplasia in GISTS arising within this syndrome, as opposed to what has been observed in *KIT* germline mutant GISTs, it has been recently hypothesized that GISTs bearing *PDGFRA* mutations, which are mutually exclusive to *KIT* mutations, may originate from telocytes instead of ICCs [46].

#### 2.2.3. KIT/PDGFRA Wild-Type GISTs

In a small portion of GISTs (around 10%), no mutation is found in either *KIT* or *PDGFRA*. These tumors can be divided into succinate dehydrogenase (SDH)-proficient and SDH-deficient [47].

Indeed, up to half of wild-type GISTs have a loss-of-function mutation or epigenetic silencing of one of the genes belonging to the SDH complex: *SDHA*, *SDHB*, *SDHC*, and *SDHD*. Regardless of the involved genes, all mutations result in the instability of the whole SDH complex, which can be identified by the loss of SDHB protein expression by immunohistochemistry [48]. The SDH complex is a key player in mitochondrial metabolism: on one hand, SDH catalyzes the conversion of succinate to fumarate within the Krebs cycle, and on the other hand, it is involved in oxidative phosphorylation, being part of the respiratory chain complex II. In SDH-deficient cells, the disruption of the above-mentioned mitochondrial functions impairs aerobic metabolism, prompting a switch to anaerobic glycolysis regardless of the presence of oxygen, a known tumorigenic mechanism known as “enhanced glycolysis”. Indeed, the accumulation of succinate leads to the upregulation of the hypoxia-inducible factor (HIF) angiogenic pathway [49].

In addition, epigenetic changes have been proposed as a tumorigenic mechanism in SDH-deficient GISTs [50,51]. Although *SDH* mutations can be somatic, the majority of SDH-deficient GISTs are syndromic and occur as a rare manifestation of pheochromocytoma–paraganglioma syndromes. The association of pheochromocytoma–paraganglioma and GISTs, the so-called Carney–Stratakis syndrome, occurs more frequently in carriers of *SDHA* and *SDHD* variants. SDH-related syndromic GISTs are often multifocal and frequently localized in the stomach [52].

SDH-proficient GISTs are a heterogeneous group characterized by different driver mutations and other genomic aberrations. In around 4% of these, the driver gene is *BRAF*, which encodes for a serine/threonine protein kinase involved in the mitogen-activated kinase (MAPK)/ERK signaling pathway [53]. BRAF activation promotes cell proliferation, differentiation, and survival [54,55]. Usually, the gain-of-function *BRAF* mutation found in GISTs is the V600E, which is a common driver of melanoma, papillary thyroid carcinoma, and several other malignant tumors [54,55]. Due to its involvement in several pathways shared with *KIT*, *BRAF* mutations result in a constitutive activation of c-KIT downstream effectors [56]. The MAPK pathway is negatively regulated by NF1, whose loss of expression is observed in a small subset of wild-type GISTs. It is almost always caused by a germline loss-of-function variant, followed by a somatic second hit and is a rare manifestation of neurofibromatosis type 1 [57,58].

With the identification of fusion genes involving the neurotrophic receptor tyrosine kinase genes (*NTRK1*, *NTRK2*, and *NTRK3*) as actionable oncogenic drivers, several clinical trials investigating their presence and role in a tissue-agnostic manner have been conducted, and most of them are still ongoing. Although *NTRK* translocations have been described in GISTs, this occurrence is rare, as fewer than 20 cases have been reported so far, to our knowledge [59,60,61,62,63,64,65]. Moreover, this assumption has been recently challenged, and therefore, as opposed to specific soft-tissue sarcoma subtypes, such as infantile fibrosarcoma, the impact of NTRK fusions as oncogenic drivers in GISTs is currently unclear [66].

### 2.3. GIST Mutational Spectrum by Site

GISTs with *KIT* exon 11 mutations can arise anywhere in the gastrointestinal trait, although more than 80% of them occur in the stomach and are frequently larger than 2 cm at the time of diagnosis. *KIT* exon 9 mutations, on the other hand, are present in up to a quarter of GISTs originating in the small intestine and in 10–15% of rectal GISTs, but are rarely found in gastric GISTs [67,68]. *KIT* exon 13 mutant GISTs are more frequently of gastric origin, whereas exon 17 mutation occurs twice as much in small bowel compared to gastric GISTs [69].

More than 90% of *PDGFRA*-mutant GISTs are of gastric origin and, especially in cases of exon 18 mutations, tend to display a prognostically favorable phenotype [70].

SHD-deficient GISTs are predominantly found in the stomach and, when occurring in syndromic cases, are often multifocal [71].

### 2.4. Secondary Mutations

As a result of both stochastic events and evolutionary pressure, subclones with additional *KIT* and *PDGFRA* mutations can develop during treatment with TKIs. For instance, KIT exon 11 mutations are almost always primary, whereas secondary mutations occur more frequently in exons 13–14 and in exons 17–18. Both types of former secondary mutations confer resistance to imatinib, the former impairing its binding to the ATP-binding domain, and the latter through changes in the activation loop, which stabilizes KIT in its active conformation Depending on the number and type of mutations, a variable response to different TKIs and therapeutic schedules has been observed [72,73]. Resistance due to mutations in exons 13 and 14 (usually the V654A and T670I aminoacidic substitutions, respectively), can be overcome by sunitinib, which retains affinity for the ATP-binding pocket. However, this molecule is not effective against exon 17 and exon 18 mutations. These latter mutations can be targeted by ripretinib and avapritinib, although, compared to exon 17, the response to avapritinib appears to be lower in exon 18 mutant tumors [25].

Secondary mutations have also been described in the *PDGFRA* gene and can involve either exon 18 outside codon 842 (activation loop) or exons 13, 14, and 15 (ATP binding domain), with repercussions in response to current therapies. For instance, specific mutations located between exons 13 and 15 co-occurring with D842V have been recently implicated in secondary resistance to avapritinib [74].

Due to BRAF involvement in the MAPK signaling pathway downstream of KIT and PDGFRA [75], BRAF secondary mutations can overcome KIT and PDGFRA blockades through an upregulation of the MAPK pathway, thereby conferring resistance to TKI in GISTs with actionable *KIT* and *PDGFRA* mutations [53,76].

## 3. State of the Art of Target Therapies for Advanced GISTs

GISTs are generally resistant to conventional chemotherapy, but the overall survival of metastatic GISTs underwent a seismic shift after the approval of imatinib in 2002.

The first-line standard treatment for metastatic GISTs is guided by their molecular characterization and analysis of the tumor’s mutational status, given the high rate of molecular alterations harbored by advanced GISTs.

The *KIT* mutational status must be evaluated in order to distinguish between imatinib-sensitive and non-sensitive mutations.

### 3.1. First-Line

#### 3.1.1. Imatinib-Sensitive Mutations

Imatinib is a selective, small molecule inhibitor of three receptor tyrosine kinases: the transmembrane receptor KIT, the chimeric BCR-ABL fusion oncoprotein of chronic myeloid leukemia, and PDGFRA. This drug was approved by the FDA in 2002 for the treatment of advanced GISTs in light of its results on 147 patients with metastatic GISTs. Overall, 79 patients (53.7%) had a partial response and 41 patients (27.9%) had stable disease [77]. The median duration of response was 29 months (95% CI, 22 to 43) at a long-term follow-up of 71 months [78]. The most common adverse events included edema (74%, frequently periorbital), nausea (52%), diarrhea (45%), myalgia or musculoskeletal pain (40%), fatigue (35%), dermatitis or rash (31%), headache (26%), and abdominal pain (26%).

For metastatic GISTs with imatinib-sensitive mutations, such as *KIT* exon 11 mutation, imatinib at a dose of 400 mg per day is the standard of care. The optimal imatinib dose to use in the first-line setting was determined in two phase III trials that randomized patients with advanced GISTs to receive imatinib at 400 mg or 800 mg daily. The EORTC 62,005 trial enrolled 946 patients. The higher imatinib dose arm yielded a statistically significant progression free survival (PFS) rate of 56% versus 50% for the lower dose arm. The overall response rates were similar between the two treatment arms. The higher dose arm was associated with greater toxicity [79]. Nevertheless, in patients with *KIT* exon 9 disease, the PFS was significantly longer in patients that received a double dose of imatinib compared to those that received the standard dose. The SWOG S0033/CALGB 15105 trial enrolled 746 patients. After a median follow-up of 4.5 years, no significant difference was observed with respect to the median PFS (18 vs. 20 months), median OS (55 vs. 51 months), and objective response rate (40% vs. 42%) between the standard dose and the higher dose arms, respectively [80]. In fact, this trial failed to demonstrate a PFS benefit favoring the *KIT* exon 9 mutant cohort that received the higher dose imatinib [78]. The metaGIST meta-analysis regarding both the EORTC 62005 and the SWOG S0033/CALGB 15105 studies confirmed a PFS advantage in favor of patients with *KIT* exon 9 mutant disease who received a double dose of imatinib (400 mg, twice daily) compared to the standard dose [81]. As a result, the standard dose for patients with *KIT* exon 9 mutant advanced GISTs receiving first-line imatinib is 800 mg daily since a higher response rate and progression-free survival (PFS) are described with a higher dose.

First-line treatment with imatinib should be continued indefinitely until disease progression or unacceptable toxicity because treatment interruption is generally followed by tumor progression regardless of the pattern of response achieved with imatinib before interruption [82,83]. Even though imatinib is highly effective in the first line, and up to 30% have remained progression-free after 5 years and 7 to 9% after 10 years [84,85], the great majority of patients will develop disease resistance and acquired secondary mutations in *KIT* or *PDGFRA*, which constitutes the main mechanism of failure to imatinib in 70 to 90% of GIST patients. In cases of tumor progression on imatinib 400 mg per day, an option may be to increase the imatinib dose to 800 mg per day orally in patients with *KIT* exon 11 mutation [86]. For *KIT* exon 9 mutant patients progressing after imatinib 800 mg daily, subsequent treatment lines must be considered.

The *PDGFRA* status must be considered for advanced GISTs as well. Patients with *PDGFRA* non-exon 18 D842V mutations are sensitive to imatinib and thus treated with this agent. Conversely, patients with *PDGFRA* exon 18 D842V mutation are resistant to imatinib.

#### 3.1.2. Imatinib Non-Sensitive Mutations: Avapritinib

*PDGFRA* D842V mutant GISTs account for 5–6% of GISTs and exhibit primary resistance to imatinib and sunitinib therapy [45]. In fact, *PDGFRA* D842V mutant GISTs are clinically distinct from imatinib-sensitive disease and often characterized by a more indolent natural history. *PDGFRA* mutations seem to be less frequent in the advanced setting, and only 11% of *PDGFRA* mutant GISTs usually develop metastasis. Moreover, first-line imatinib was shown to be ineffective in this molecular subtype since most patients treated with imatinib underwent disease progression at the first radiographic assessment, and the ones who experienced long disease stability had a lower mitotic count, and thus, more indolent intrinsic behavior [87]. Until January 2020, no standard systemic therapy was available for this GIST molecular subtype, and surgical resection was the preferred option for oligoprogressive disease. Nowadays, 300 mg of avapritinib per day orally should be considered as the preferred option for these patients given the impressive response rate (ORR: 91%) and the duration of response (median DOR: 27.6 months) provided in the NAVIGATOR trial [88]. Avapritinib is a highly selective oral inhibitor of *PDGFRα* mutant kinases. The NAVIGATOR phase I trial enrolled 231 patients with advanced GISTs. Among these patients, 56 had *PDGFRA* exon 18 D842V mutant disease (24%), 8 had non-D842V mutant disease (4%), and 167 had *KIT* mutant disease (72%). In the *PDGFRA* D842V-mutant population, the ORR was 91% and the disease control rate was 98% (CR = 13%, PR = 79%, SD = 6%), which is the best achievement ever reached for this GIST molecular subtype [89]. For these reasons, avapritinib was approved by the FDA in January 2020 in the first-line setting for adults with advanced *PDGFRA* exon 18 mutant GISTs and by EMA in 2022 only for *PDGFRA* D842V mutant patients. Based on these promising first-line results, the VOYAGER trial was conducted, randomizing patients to avapritinib versus regorafenib as third-line or further treatment independently of their KIT and PDGFRA mutational status. This trial showed no statistically significant difference in the median PFS between avapritinib and regorafenib in patients with unselected late-line GISTs. The patients with *PDGFRA* D842V mutations demonstrated significantly higher PFS in the avapritinib arm (median PFS not reached) compared to the regorafenib arm (median PFS of 4.5 months) [90].

#### 3.1.3. Imatinib Non-Sensitive Mutations: Larotrectinib, Entrectinib, BRAF Inhibitors

For patients with *KIT/PDGFRA* wild-type GISTs, their *SDH, BRAF*, and *NTRK* gene statuses should be evaluated in order to identify the best patient-tailored approach in terms of treatments and follow-up. For GISTs with *NTRK* rearrangements, larotrectinib and entrectinib can be considered due to their tumor agnostic approval in Europe for unresectable locally advanced or metastatic GISTs [60,91,92]. In fact, a pooled analysis of three phase I or II clinical trials enrolling 153 patients with *NTRK* rearranged solid tumors showed that four patients with GISTs treated with larotrectinib showed an objective response rate of about 80% [62]. In a further pooled analysis of the same three phase I or II clinical trials, three patients with GISTs had *ETV6-NTRK3* gene fusions, and one of them experienced complete response [93]. In an integrated analysis of two phase I and one phase II trials, one patient enrolled with a GIST showed a partial response as the best response with entrectinib [94]. BRAF inhibitors and BRAF-MEK inhibitor combinations should be considered an off-label indication justified by biological plausibility in *BRAF*-mutated patients [95]. In fact, BRAF inhibitors have demonstrated antitumor activity in clinical trials on patients with BRAF mutant malignancies, such as melanoma, and one case report described a case of a GIST that experienced tumor regression with dabrafenib [96].

#### 3.1.4. Imatinib Non-Sensitive Mutations: Sunitinib and Regorafenib

SDH-deficient GISTs could benefit from alternative therapeutic approaches. For example, primary surgical debulking or any interval surgery can be considered for this GIST subtype and, in particular, for SDHA-deficient GISTs. In fact, the indolent nature of these tumors makes them more suitable for a primary surgical debulking or any interval surgery when limited progression is observed [71]. For SDH-deficient GISTs with progressive disease, sunitinib and regorafenib were reported to be active and should be considered as the standard first-line option [97].

### 3.2. Second-Line Treatment

#### Sunitinib

For progressive GISTs, 50 mg of sunitinib daily following a “4 weeks on–2 weeks off” regimen is the standard second-line treatment. This agent was approved given its efficacy in terms of a PFS prolongation of 5.8 months. As described for imatinib, the *KIT* mutational status correlates with sunitinib’s efficacy. In fact, among the primary *KIT* mutations, exon 9 mutant GISTs are known to have higher response rates compared to *KIT* exon 11 mutant GISTs. Regarding *KIT* secondary mutations, exon 13 or 14 mutant GISTs showed longer PFS and OS compared to *KIT* exon 17 or 18 mutant GISTs [98].

Sunitinib on a different schedule with a lower daily dose of 37.5 mg continuously is well-tolerated and effective, even if no formal comparison has been made in any randomized clinical trial. The most common drug-related adverse events were hypertension, diarrhea, and hypothyroidism [99].

### 3.3. Third-Line

#### Regorafenib

After progression to sunitinib, regorafenib at a dose of 160 mg daily for 3 weeks, followed by a 1 week pause, should be considered as a successive treatment option, with a significant improvement in PFS. In fact, in the randomized phase III trial that tested regorafenib versus the placebo after imatinib and sunitinib failure, a 4.8 month increase in PFS was described for regorafenib compared to 0.9 months of mPFS for the placebo [100]. The 98% of patients receiving regorafenib experienced drug-related adverse events, versus 68% patients in the arm assigned to the placebo. The most frequent grade 3 or higher adverse events in the regorafenib arm were hypertension (31 of 132, 23%), hand–foot skin reaction (26 of 132, 20%), and diarrhea (7 of 132, 5%). In addition, personalized schedules of regorafenib demonstrated to improve therapeutic outcomes and are commonly adopted in clinical practice in high-volume GIST centers to maximize long-term therapy [101].

### 3.4. Beyond Third-Line

#### 3.4.1. Ripretinib

For pretreated GISTs, ripretinib can be considered a further treatment option. Ripretinib is a switch-control TKI with a dual mechanism of action that provides broad-spectrum inhibition of KIT and PDGFRA activity. Resistance to the approved inhibitors of *KIT* proto-oncogene, receptor tyrosine kinase (*KIT*), and platelet-derived growth factor receptor α (PDGFRA) is a serious clinical challenge for patients with advanced gastrointestinal stromal tumors. In the prospective, randomized, phase III INVICTUS clinical trial, 150 mg of ripretinib once daily was compared to a placebo in patients who progressed after at least two treatment lines with tyrosine–kinase inhibitors. The median progression-free survival was 6.3 months with ripretinib compared to 1.0 months with the placebo (hazard ratio of 0.15, 95% CI 0.09–0.25; *p* < 0.0001), and the median OS was 15.1 months versus 6.6 months. The ripretinib ORR < 10% was in line with that of prior TKIs, and most of its clinical benefit derived from disease stabilization in 47% of patients at week 12. Treatment-related serious adverse events were described in 9% of patients in the ripretinib arm and in 7% of patients in the placebo arm. The most common of these were lipase increase, hypertension, fatigue, and hypophosphataemia. This agent was thus approved by EMA and can thus be considered a fourth-line treatment option in patients with advanced GISTs [102]. For instance, whether a double dose of ripretinib (150 mg BID) extends the effect of this drug as it does with imatinib in the first-line setting was the question asked in a phase I trial that dose-escalated patients who experienced disease progression at a 150 mg to a 150 mg twice-daily dose [103]. In this trial, the ripretinib dose escalation provided continued clinical benefit in advanced GISTs across second, third, and later lines, with an mPFS of 5.6, 3.3, and 4.6 months for patients on second-, third-, and ≥fourth-line therapies, respectively.

Ripretinib was also tested in a phase I trial on GIST patients with imatinib-resistant secondary *KIT* mutations, showing a median PFS in the second-line setting of 10.7 months [104]. Ripretinib was thus compared to sunitinib in the phase III INTRIGUE trial in patients who were previously treated with imatinib. In this trial, ripretinib was not superior to sunitinib in terms of PFS in patients previously treated with imatinib. However, the median PFS observed with ripretinib was comparable to the median PFS observed, with sunitinib suggesting that ripretinib is active as the second-line therapy for GISTs. Additionally, the ORR in *KIT* exon 11 patients treated with ripretinib was higher compared to the ORR in patients treated with sunitinib. Ripretinib also demonstrated a more favorable safety profile [105]. Moreover, recently published data from the INSIGHT trial demonstrated that next-generation sequencing (NGS) analysis of participants’ circulating tumor DNA (ctDNA) revealed distinct molecular subgroups that strongly benefited from ripretinib or sunitinib. In fact, patients with imatinib-resistant mutations in the KIT-activating loop (exons 17/18) derived meaningful clinical benefit from ripretinib but not sunitinib, whereas patients with imatinib-resistant mutations in the KIT ATP-binding pocket (exons 13/14) derived meaningful clinical benefit from sunitinib but not ripretinib. This study demonstrated the potential value of ctDNA NGS-based sequencing of *KIT* to predict the clinical benefit of ripretinib or sunitinib as second-line therapy in patients with advanced GISTs [106].

#### 3.4.2. Imatinib Rechallenge

According to Kang et al. (RIGHT study), for patients with heavily pretreated metastatic GISTs who experience disease progression on other TKI treatment, rechallenge with imatinib may improve patients’ outcomes. In fact, imatinib resumption compared to placebo improved progression-free survival (1.8 vs. 0.9 months, 95% CI, 0.9–1.7) and the disease control rate at 12 weeks (32% vs. 5%) [107].

#### 3.4.3. Cabozantinib

According to the NCCN guidelines, cabozantinib can also be considered as a further treatment line as a result of the CaboGIST EORTC 1317 trial, which showed an ORR of 14% with an encouraging disease control rate of 82% and a median PFS of 5.5 months. Among the patients enrolled, 30 of 50 were progression-free after 12 weeks [108].

#### 3.4.4. Dasatinib, Pazopanib, and Everolimus Plus TKI

According to the NCCN guidelines, dasatinib, pazopanib, and everolimus plus TKI can also be considered for the treatment of advanced GISTs [109,110].

A treatment strategy proposal with the above-mentioned drugs is illustrated in the following figure (Figure 1).

## 4. Future Perspectives and Ongoing Clinical Trials

As previously discussed, the treatment of advanced GISTs may present some weaknesses due to primary or secondary resistance phenomena, treatment intolerance, or treatment failure caused by disease progression. Therefore, novel therapeutic approaches are needed to overcome these limitations.

The understanding of GIST biology has helped to develop new therapeutical strategies in the advanced setting. In Table 1, the ongoing clinical trials for molecularly selected or unselected advanced GISTs in different treatment lines are summarized.

### 4.1. TKI-Based Strategy

The mechanisms of resistance to TKIs are generally represented by a polyclonal expansion of heterogeneous subpopulations of cells harboring different KIT or PDGFRA mutations [111]. Several efforts have been made to find novel compounds capable of targeting a broader range of resistance mutations.

As reported in Table 1, many phase I or phase II trials are investigating the role of TKIs, alone or in combination with other classes of drugs. However, the combination of targeted therapies, with the purpose of increasing the treatment response and prolonging it, usually carries the risk of enhancing toxicities. One of the alternative solutions investigated to reduce toxicity was a rapid drug substitution (three days of sunitinib followed by four days of regorafenib over a 28-day cycle) in “TKI-refractory” patients, with the aim of increasing tolerability and trying to keep the response rate [112].

*KIT* and *PDGFRA* mutations are associated with the activation of RAS/RAF/MAPK, JAK/STAT3, and PI3K/AKT/mTOR downstream pathways, increasing cell proliferation and inhibiting apoptosis [113].

Combining the inhibition of KIT/PDGFRA with targets such as RAS/MAPK and PI3K/mTOR pathway, as a promising alternative to enhance apoptosis, may have the potential rationale to overcome standard TKI failure [114]. Moreover, a phase Ib or II trial will explore the association of imatinib 400 mg daily with the MEK inhibitor “MEK162”, both in locally advanced or metastatic GISTs and in newly diagnosed treatment-naïve GISTs (NCT01991379).

### 4.2. DNA Damage Repair (Temozolomide)

In 2003, Trent and colleagues reported the results of a phase II trial of temozolomide in subsequent treatment lines. The aim of the study was to determine the efficacy and tolerability of temozolomide in soft tissue sarcomas (arm 1) or in patients with GISTs (arm 2). Temozolomide was demonstrated to be well tolerated, but only minimal efficacy was reported in molecularly unselected GIST patients [115].

SDH-deficient GISTs are characterized by global DNA hypermethylation. This phenotype causes a prevalence of O6-methylguanine-DNA methyltransferase (MGMT) promoter methylation and may imply sensitivity to alkylating agents [47]. The alkylating agent temozolomide induces cytotoxic damage through DNA methylation at the O6 position of the DNA base guanine, resulting in apoptosis and tumor cell death [116]. A loss of tumoral MGMT expression, as shown in SDH-deficient tumors, was associated with sensitivity to temozolomide in other types of tumors [116]. Based on this rationale, a phase II clinical trial of temozolomide monotherapy in SDH-mutant/deficient GISTs is currently ongoing (NCT03556384).

### 4.3. Antibody–Drug Conjugates (ADCs)

ADCs are highly targeted agents that combine monoclonal antibodies specific to cellular-surface antigens with a biologically active cytotoxic payload or drug [117]. Several ADCs have become part of clinical practice in different types of neoplasm, such as breast cancer and urothelial carcinoma. GPR20 is an orphan G protein-coupled receptor selectively expressed in more than 80% of all GISTs, regardless of the molecular subtype. DS-6157 is an investigational GPR20-targeting ADC with a tetrapeptide-based linker and DNA topoisomerase I inhibitor exatecan derivative (DXd). Preclinical pharmacokinetics and a safety profile supported the clinical development of DS-6157 as a novel GIST therapy for patients refractory or resistant to approved TKIs [118]. A phase I trial assessed the safety, efficacy, and pharmacokinetics of DS-6157a in advanced GISTs (NCT04276415).

Preclinical studies have also investigated the role of NN2101, a fully human IgG1 conjugated with DM1 (a microtubule inhibitor), reporting antitumor activity in various cancer cell lines. In mouse xenograft models, the combination of NN2101-DM1 induced complete disease remissions, suggesting a potential role for wild-type and c-kit-positive cancers [119].

### 4.4. Immunotherapy

The role of immunotherapy in GISTs is still unclear, although the mechanisms of immune escape and immune surveillance are known to be well represented in this kind of tumor [120].

Particularly, the immune tumor microenvironment of GISTs seems to be highly infiltrated with immune cells. In addition, immune-escape mechanisms are widely represented, with immunosuppressive M2 macrophages, the over-expression of indoleamine 2,3-dioxygenase (IDO) or PD-L1, and the loss of major histocompatibility complex type 1 [121].

The anti-PD-(L)1 antibody nivolumab, alone or in combination with anti-CTLA-4 ipilimumab, failed to demonstrate activity in GISTs [122]. Likewise, in the PEMBROSARC trial, pembrolizumab, in combination with metronomic cyclophosphamide in soft tissue sarcomas, failed to demonstrate responses in the subgroup of patients with advanced GISTs [123].

Given that, a rationale to combine immune checkpoint inhibitors with other drugs was explored.

In a phase Ib trial, a combination treatment with anti-CTLA4 ipilimumab and multi-TKI dasatinib did not demonstrate any efficacy [124]. Moreover, a phase II trial investigated the association of pembrolizumab and axitinib in several soft-tissue sarcomas, including GISTs (NCT02636725).

Seifert et al. reported that imatinib treatment in GISTs could enhance the expression of PD1 on T-cells, promoting T-cell activity and, consequently, providing a rationale for combining imatinib and anti-PD1/PDL-1 drugs [125]. For this reason, ongoing trials are exploring the combination of anti-PD(L)1 and imatinib (NCT05152472/NCT03609424/NCT01738139) or other TKIs, such as regorafenib (NCT03475953).

Another class of immunomodulatory drugs is represented by cytokines, a group of molecules that can regulate innate and adaptive immunity. Cytokine therapy can switch the balance in favor of increasing immune surveillance, thereby eradicating drug-resistant clones [126]. Based on this rationale, Chen et al. combined peg-interferon alfa 2b, which promotes Th1 response, with imatinib; a total of 8 patients with stage III or IV GISTs were enrolled, and after a follow-up of 3.6 years, complete response or partial response was seen in 100% of the patients [127].

Moreover, preclinical studies have shown that VEGF inhibitors may have immunomodulatory effects. Indeed, VEGF can lead to immunosuppression by inhibiting dendritic cell maturation, promoting the infiltration of immune suppressive cells, and enhancing immune checkpoint molecule expression [128]. As VEGF and checkpoint inhibitors act in different steps of the immune response, their dual blockade may have a synergistic effect, and the combination of anti-VEGF therapy with avelumab, an anti-PD-L1 antibody, is under investigation (NCT04258956).

Among others, one of the roles of constitutional c-kit activation is represented by the induction of the overexpression of indoleamine 2,3-dioxygenase (IDO), whose function is fulfilled by the metabolization of amino acid tryptophane, which promotes a tolerogenic tumor microenvironment [129]. In the phase II trial PEMBROSARC, IDO was overexpressed in 63% of the patients pretreated with imatinib [130]. To date, the combination of immunotherapy and IDO inhibitors in melanoma patients has been quite disappointing, but the rationale for targeting IDO in GISTs is of great interest [130]. The combined inhibition of PDL1 and IDO was also explored in a phase II study with pembrolizumab and epacadostat in patients with imatinib refractory advanced GISTs (NCT03291054).

### 4.5. Radioligand Therapy

One of the most promising developments of the last decade is constituted by radiolabeled peptides, a group of molecules that can be exploited for diagnostic and therapeutic purposes. In prostate cancer, radium-223 and 177Lu-PSMA therapy have become part of clinical practice in metastatic castration-resistance settings [131]. One of the main targets identified for GISTs patients is the gastrin-releasing peptide receptor (GRPR), which is usually overexpressed in GIST cells [132]. NeoB is a radiotracer designed for targeting the GRPR that can be employed for diagnosis or for treatment, with which the radioisotope is coupled [133]. A phase I/IIa trial evaluated the role of 68Ga-labelled antagonists to GRPRs as radioligands for nuclear imaging in GISTs. 68Ga-NeoBOMB1-PET showed variable uptakes in advanced-GISTs, resulting in a possible new diagnostic option in selected cases [134]. Currently, a phase I/IIa trial is testing 177Lu-NeoB to confirm its potential as a theragnostic agent in adult patients with advanced solid tumors with an overexpression of GRPR (NCT03872778).

### 4.6. Other Options

Several classes of drugs are under investigation to determine their activity, efficacy, and tolerability for different tumors, including advanced GISTs.

Histone deacetylase inhibitors (HDACis), with their involvement in the regulation of gene expression and chromatin remodeling, seemed to be valid anti-cancer molecules. The pan-deacetylase inhibitor LBH589 achieved FDA approval in multiple myelomas and may be a treatment option for patients with GISTs as well [135].

The molecular chaperone heat shock protein 90 (HSP90) is required for the proper folding and stabilization of KIT and PDGFRA. Both imatinib-sensitive and imatinib-resistant GIST cell lines are inhibited in growth and apoptosis by pimitespib, an HSP90 inhibitor [136]. The phase III trial CHAPTER-GIST-301 enrolled patients with TKI-refractory GISTs to receive oral pimitespib or placebo. The results were quite promising, as pimitespib improved PFS and cross-over-adjusted OS. Exploratory analyses demonstrated clinical benefit irrespective of the *KIT* mutational status [137].

Another treatment option explored was the CDK 4/6 inhibitor ribociclib. The role of the oncogenic activation of CDK 4/6 was linked to cell cycle progression. CDK 4/6 inhibitors as monotherapy failed to demonstrate clinical activity in patients with GISTs. Therefore, a preclinical study suggested the role of combined CDK2 and CDK 4/6 inhibitors to maximize responses in GIST patients [138].

To overcome treatment resistance, a multicenter phase Ib study is testing the combination of ribociclib with spartalizumab (an anti-PD1 molecule) in patients with several malignancies, including GISTs (NCT04000529).

## 5. Conclusions

GISTs are malignant neoplasms that have had one of the greatest improvements in survival during the last twenty years. This has mainly been due to the discovery of the driver mutation of *KIT*, and subsequently, the other less common molecular alterations that lead to the development of specific targeted drugs.

Nowadays, the molecular characterization of GISTs is needed in every setting because it has implications for the treatment options and the perspective of a possible genetic syndrome for wild-type GISTs.

Nevertheless, the more we can administer new drugs targeting specific molecular alterations, the more we see new resistances arising that need to be studied.

Among the different treatment strategies, immunotherapy has not shown any strong signal to date, at least not for the all-comers population, and the current research direction is aimed at developing new drugs targeting the full spectrum of *KIT* mutations maximizing the inhibition of this oncogene pathway. Regarding non-KIT-driven GISTs, other specific drugs are under development, but also in these diseases, the direction is toward the inhibition of the specific molecular alteration with a TKI-based strategy.

For pan wild-type GISTs, even though they are becoming less frequent with the detection of new molecular alterations, the current treatment options are not very effective, and there is no ongoing trial for this specific population. Therefore, a huge international effort has to be made to better identify and treat this drug-orphaned population.

Preclinical and clinical research is ongoing to overcome resistances and possibly treat our patients with innovative strategies to maximize the great benefit already reached.

## Figures and Tables

**Figure 1 cancers-15-02074-f001:**
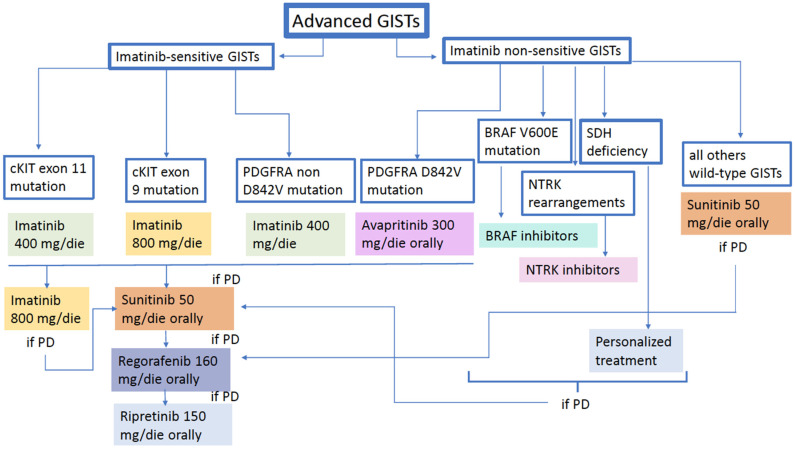
GIST treatment algorithm.

**Table 1 cancers-15-02074-t001:** Active clinical trials on GISTs.

Clinical Trial	Phase	Patient Population	Mutations	Treatment Arm (s)
NCT01991379 *	1b-2	Advanced GIST Phase Ib: locally advanced or metastatic GISTs and have progressed on imatinib. Phase II: newly diagnosed or treatment-naïve, or have been off adjuvant imatinib therapy for at least 3 months.	Not required.	Imatinib 400 mg/die + t
NCT05354388	Observational	Recurrent and/or metastatic advanced unresectable GIST. Subjects must have progressed on imatinib or have documented intolerance to imatinib.	Exclusion criteria: *PDGFRA* exon 18 mutation (including D842V).	Ripretinib 150 mg QD + surgery (if PR or SD achieved)
NCT05132738	Single-arm, single-center, exploratory study	Potentially resectable locally advanced or recurrent metastatic GIST after failure of treatment with imatinib.	Immunohistochemical detection of CD117 and/or DOG-1 positive.	Ripretinib 150 mg QD + surgery
NCT03594422	1	GIST or other solid tumor. GIST patients must be primarily resistant to imatinib (tumor progresses within 6 months of first-line imatinib treatment, or succinate dehydrogenase B (SDHB)-deficient confirmed by immunohistochemistry, or *NF1* mutation), OR imatinib or imatinib and at least one other TKI treatment failure (after imatinib or other TKI treatment for more than 6 months, tumor progresses again after achieving tumor remission or stability).	Succinate dehydrogenase B (SDHB)-deficient confirmed by immunohistochemistry, or *NF1* mutation.	HQP1351(BCR-ABL inhibitor)
NCT05152472	2	Unresectable advanced GIST after the failure of standard treatments (imatinib, sunitinib, and regorafenib).	*KIT* (exon 11) mutational status: wild-type or mutated. Exclusion criteria: D842V mutation in exon 18 of *PDGFRA*.	Imatinib 400 mg + atezolizumab 1200 mg
NCT02638766 (REGISTRI)	2	Metastatic and/or unresectable GIST previously untreated.	*KIT/PDGFR* wild-type (exons 11, 9, 13, and 17 in *KIT* gene and in 12 and 18 exons of *PDGFR* gene)	Regorafenib 160 mg QD, 3 weeks ON, 1 week OFF
NCT04193553 (LENVAGIST)	2	Unresectable advanced GIST after the failure of standard treatments (imatinib, sunitinib).	Exclusion criteria: documented mutation in *PDGFRA* exon 18 (D842V substitution).	Lenvatinib
NCT03609424	1–2	Unresectable advanced GIST after the failure of standard treatments (imatinib, sunitinib, and regorafenib).	CD117(+), DOG-1(+), or mutation in KIT or *PDGFRA* gene.	Imatinib 400 mg QD + PDR001 (anti-PD-1)
NCT03556384	2	Advanced or metastatic GIST.	SDH-mutant/deficient.	Temozolomide 85 mg/m^2^
NCT05245968 (CHAPTER-GIST-101)	1	Advanced GIST progressed during or within 6 months of the last imatinib administration at enrollment.	Not required.	Primtespib + imatinib Or Primtespib followed by imatinib Or Sunitinib
NCT05489237	1	Metastatic and/or surgically unresectable GIST, after failure of at least imatinib.	Documented pathogenic mutation in *KIT* OR any *PDGFRA* mutation other than exon 18 mutations.	IDRX-42 (small TKI)
NCT04409223	3	Advanced GIST after failure of imatinib.	Not required.	Famitinib vs. sunitinib
NCT04258956 (AXAGIST)	2	Advanced GIST (no more than 3 previous lines of treatment, which must include imatinib and sunitinib).	Known mutational status *KIT* or *PDGFRA* (patients with *PDGFRA* D842V mutations are not eligible for this study).	Avelumab + axitinib
NCT05208047	3	Advanced GIST (documented disease progression on or intolerance to imatinib).	Exclusion criteria: known *PDGFR* driving mutations or known succinate dehydrogenase deficiency.	CGT9486 + sunitinib vs. sunitinib
NCT05160168	1–2	Advanced GIST. Cohort 1: those who have progressed on or are intolerant to imatinib, sunitinib, regorafenib, and ripretinib (≥5th line). Cohort 2: those who have progressed on or are intolerant to imatinib, sunitinib, and 0–1 additional lines of therapy in the advanced/metastatic setting (3rd–4th line). Cohort 3: those who have progressed on or are intolerant to imatinib (including in the adjuvant setting) and who have not received additional systemic therapy for advanced GIST (2nd line).	Exclusion criteria: patients known to be both *KIT* and *PDGFRA* wild-type.	THE-630 (orally administered TKI)
NCT05461664*	Observational	Metastatic or unresectable advanced GIST.	Non-exon 18 mutation of *PDGFRA*.	Avapritinib
NCT04595747	2	Sarcoma with a change in a group of proteins called fibroblast growth factor receptors (FGFRs) or SDH-deficient gastrointestinal stromal tumor (GIST).	SDH-deficient GIST regardless of FGFR status.	Rogaratinib (BAY 1163877)
NCT03475953	1–2	Advanced solid tumors (including GIST).	Not required.	Regorafenib + avelumab
NCT01738139	1	Advanced solid tumors (including GIST).	Not required.	Imatinib + ipilimumab
NCT05751733	Randomized, single-center	Previous first-line TKI (imatinib/avatinib) therapy and eventual treatment failure (disease progression or toxicity intolerance during treatment).	Not required.	Apatinib
NCT03944304	2	Patients who failed to at least imatinib, sunitinib, and regorafenib (disease progression and/or intolerance).	Histologically confirmed metastatic or unresectable GIST with CD117(+), DOG-1(+), or mutation in *KIT* or *PDGFRA* gene.	Paclitaxel

* Not yet recruiting. List of abbreviations: GIST: gastrointestinal stromal tumor; PDGFRA: platelet-derived growth factor receptor alpha; TKI: tyrosine–kinase inhibitor; NF1: neurofibromin 1; SDH: succinate dehydrogenase.

## Data Availability

No new data were created or analyzed in this study. Data sharing is not applicable to this article.

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
