# Peer review of "Molecular Tailored Therapeutic Options for Advanced Gastrointestinal Stromal Tumors (GISTs): Current Practice and Future Perspectives"

_cancers, 2023, doi:10.3390/cancers15072074_

Round 1

Reviewer 1 Report

The manuscript titled "Molecular tailored therapeutic options for advanced Gastrointestinal Stromal Tumors (GISTs): current practice and future perspectives" is, in my opinion, a well-written state-of-art of current practice in GIST. Moreover, the proposed paper is readily understandable because it is well constructed, clear, and well-described with figures exhaustive and appropriate to the subject matter. Furthermore, the paper is well supported by a careful reading of the literature in the field and provided an interesting multidisciplinary approach. 

I strongly suggest a careful reading of the text for the presence of several typos and errors. 

Author Response

Thank for your comment and for the appreciation of our work.

We corrected the typos and errors signaled.

Moreover we improved the English style.

Reviewer 2 Report

Article Molecular tailored therapeutic options for advanced Gastrointestinal

Stromal Tumors (GISTs): current practice and future perspectives is

well thought out and  written.  The authors described in the paper the methods of GIST treatment being developed. The work on these methods is still ongoing, and they may be used in clinical practice in the future.

Gastrointestinal stromal tumors (GISTs) are common mesenchymal tumors characterized by various molecular changes manifest different clinical presentations and behaviors.

Understanding   of these mutations gives many

new treatment options .

There are new drugs that have proven effective in GIST and

this review explores their molecular-related mechanisms of action

paths characteristic of this disease.

This review provides a overview of the GIST and molecular pathways

their respective tailored therapeutic strategies.

The future prospects for new treatment options and drugs

are analyzed and discussed. The introduction is well prepared and supported by literature. The division of the text into subsections makes orientation easier.

The list of treatment trials in the table is clear.

Although much progress has been made in understanding the origin of GISTs, more effective treatment strategies are still needed. The authors of the publication synthetically describe the treatment options for GISTs and look for future directions of treatment that may be effective

Author Response

Thank you for your comment.

We performed the editing of English language and style as requested.

Reviewer 3 Report

The current review responds to a large extent to the issues it analyzes. Below are some comments for its further improvement.
1. "Most GISTs occur sporadically among middle-aged adults (median age of 60-65 years) with a male predominance [4]" This information concerns 929 patients. There should be a reference on a larger scale of patients.
2. 2.1 more biomarkers should be mentioned
3. 2.2 the 24th reference would be better to mention an important and recent review, not a clinical trial. Also, the rest of the references in the section should be more recent.
4. 2.2.2 and 3.1.2 The D842V mutation is of particular interest and may be further developed
5. 2.4 resistance to treatments may be further analyzed
6. 3.1.3 and 3.1.4, are there clinical trials for these drugs?
7. Regarding section 4, extensive work has been made on the new treatments, but others have not been mentioned. Also, there are many more clinical studies specifically in immunotherapy.
8. It would be good to make a scheme that relates the new treatments to the responsible mutation
9. The conclusion is very general. Which treatments seem more effective and why should they be given more focus?

Author Response

Thank you for your nice and very useful comment.

Here the point-by-point response:

  1. "Most GISTs occur sporadically among middle-aged adults (median age of 60-65 years) with a male predominance [4]" This information concerns 929 patients. There should be a reference on a larger scale of patients. We added a more appropriate reference [Søreide K, Sandvik OM, Søreide JA et al. Global epidemiology of gastrointestinal stromal tumours (GIST): A systematic review of population-based cohort studies. Cancer Epidemiol. 2016 Feb;40:39-46. doi: 10.1016/j.canep.2015.10.031. Epub 2015 Nov 24. PMID: 26618334]
    2. 2.1 more biomarkers should be mentioned.
    We mentioned more biomarkers such as CD34, SMA, s100 and desmin
    3. 2.2 the 24th reference would be better to mention an important and recent review, not a clinical trial. Also, the rest of the references in the section should be more recent. We added more recent references including the one referring to the recent review as requested.
    4. 2.2.2 and 3.1.2 The D842V mutation is of particular interest and may be further developed.
    We deepened the explanation regarding this mutation both in the molecular characterization part and in the therapeutical one.
    5. 2.4 resistance to treatments may be further analyzed. We further developed the secondary resistance paragraph.
  2. 3.1.3 and 3.1.4, are there clinical trials for these drugs? There are no trials ongoing on these drugs.
    7. Regarding section 4, extensive work has been made on the new treatments, but others have not been mentioned. Also, there are many more clinical studies specifically in immunotherapy. We added more clinical studies in immunotherapy as requested.
    8. It would be good to make a scheme that relates the new treatments to the responsible mutation. We did not make a new scheme as it would be too wide and too complicated. Moreover, there would be a lot of overlapping treatment strategies for the same line/molecular subtype.
    9. The conclusion is very general. Which treatments seem more effective and why should they be given more focus? We adjusted our conclusion as requested.

Moreover, we edited the English style and corrected the typos and errors.

Round 2

Reviewer 3 Report

The paper can be further published in the present form